# Bioresponsive Polymers for Nanomedicine—Expectations and Reality!

**DOI:** 10.3390/polym14173659

**Published:** 2022-09-03

**Authors:** Sabina Quader, Joachim F. R. Van Guyse

**Affiliations:** 1Innovation Center of NanoMedicine, Kawasaki Institute of Industrial Promotion, 3-25-14 Tonomachi, Kawasaki-ku, Kawasaki 212-0821, Japan; 2Leiden Academic Center for Drug Research (LACDR), Leiden University, 2333 CC Leiden, The Netherlands

**Keywords:** nanomedicine, targeted therapy, stimuli-sensitive, pH-sensitive, thermoresponsive, enzyme-responsive, ROS-sensitive, glutathione-responsive, ATP-responsive

## Abstract

Bioresponsive polymers in nanomedicine have been widely perceived to selectively activate the therapeutic function of nanomedicine at diseased or pathological sites, while sparing their healthy counterparts. This idea can be described as an advanced version of Paul Ehrlich’s magic bullet concept. From that perspective, the inherent anomalies or malfunction of the pathological sites are generally targeted to allow the selective activation or sensory function of nanomedicine. Nonetheless, while the primary goals and expectations in developing bioresponsive polymers are to elicit exclusive selectivity of therapeutic action at diseased sites, this remains difficult to achieve in practice. Numerous research efforts have been undertaken, and are ongoing, to tackle this fine-tuning. This review provides a brief introduction to key stimuli with biological relevance commonly featured in the design of bioresponsive polymers, which serves as a platform for critical discussion, and identifies the gap between expectations and current reality.

## 1. Introduction

Nanomedicine (NM) has already been established as a critical and enabling technology, with more than 50 commercial formulations and over 400 in clinical development stages, potentially transforming treatment outcomes for millions of patients worldwide [1]. Among the NM formulations currently in clinics or clinical trials, many are polymer-based NMs, such as polymeric drugs, polymer–drug conjugates, polymeric nanoparticles, polymer–protein conjugates, polymeric micelles, dendrimers, nanogels, and polyplexes, which are used to treat a wide range of clinical conditions, including cancer, inflammation, autoimmune diseases, and neurological disorders [2,3,4,5]. The extensive variety of polymeric materials, their diverse biophysical/biochemical properties with a practical scope for further modulation, ease of synthesis and manufacturing, and, in many instances, excellent biocompatibility, present compelling arguments for their use in biomedical applications. The incorporation of a bioresponsive or sensory function, i.e., the phenomenon of changing properties in response to biological cues, such as temperature, pH, reductive environment, glucose, ATP concentration, and tissue-specific enzymes, is also achievable in polymer platforms [6] due to their controllable molecular architecture.

The fundamental appreciation of bioresponsive polymers for NM applications is that the therapeutic function of NM can be selectively activated at the diseased or pathological site; a fundamental concept towards developing targeted therapy. In this process, the selective activation of NMs, which primarily entails the release of therapeutic payloads, is achieved by exploiting the inherent abnormality or malfunction of the pathological sites (Figure 1). For example, compared to normal tissues, the inflamed tissue-associated pathological milieu (oxidative stress, acidic pH, and overexpressed enzymes) can provide vital biochemical stimuli for the activation of NMs [7]. Again, tumor tissues have a different microenvironment than normal tissues in terms of pH, redox conditions, and enzyme expression, which can be exploited to design tumor-sensitive NMs [8]. In these cases, NMs benefit from their spontaneous accumulation in tumors, due to the tumor-associated leaky vasculature. This spontaneous accumulation of NM is a common anatomical and pathophysiological characteristic of solid tumors that is commonly known as the “enhanced permeability and retention” (EPR) effect [9]. Although, the primary aims and expectations in designing bioresponsive polymers are to gain exclusive selectivity of therapeutic action in diseased sites, whilst sparing the healthy tissue, this remains particularly challenging to realize. One of the main struggles in achieving this desired selectivity is that the characteristics distinguishing pathological sites from healthy tissue are diffuse, differing only slightly or even overlapping. Consequently, refined strategies are required to fine-tune the polymer structure to the pathological site of interest [10]. Various approaches have been adopted to accomplish this fine-tuning, such as using polymers with inherent responsive characters, installing responsive segments or linkers, and associated dynamic covalent chemistry, inducing secondary structures and related aggregation or self-assembly behavior, adopting supramolecular assembly, or a combination of these strategies. Additionally, a combination of responsive stimuli [11], such as pH–redox-sensitive [12], pH–thermosensitive [13,14], enzyme–thermoresponsive [15], and ATP–redox [16] dual-sensitive polymeric materials, have been developed to realize NMs that respond to subtle microenvironmental change in pathologic conditions. This review article summarizes several standard stimuli with biological relevance that are commonly considered in the design of bioresponsive polymers, which serves as a platform for critical discussion and highlights the gap between expectation and current reality.

## 2. pH-Sensitive Systems

pH is an essential physiological signal, and plays a crucial role in maintaining cell and tissue homeostasis. In healthy physiological conditions, the intra- and extracellular pH in the body ranges from 7.0–7.4, which is the ideal pH for many crucial biological processes, such as oxygenation of the blood or maintaining functional protein conformation [17]. The pH of intracellular organelles, such as endosomes, lysosomes, and the Golgi apparatus, also plays a critical role in fundamental cellular processes, such as vesicle trafficking and fusion, receptor–ligand interactions and related signaling, lysosomal degradation and autophagy, and post-translational modification of proteins and lipids (e.g., glycosylation) [7,8,9]. Specifically, endosomal pH affects the fate of plasma membrane proteins, lipids, and extracellular signals, including growth factor receptors and their ligands, by carefully controlling the sorting and trafficking of vesicular cargo for lysis or recycling. In this process, vesicles formed at the plasma membrane, where the external environment has a pH of 7.4, are initially transported or fuse to form early endosomes (pH 6.0–6.5). Early endosomes may subsequently be segregated into recycling-endosomes, which are typically more alkaline (pH 6.4–6.5). Early endosomes that mature into late endosomes, on the other hand, are often more acidic (pH 5.5–6.0). After completing the proteolysis process and undergoing further acidification, late endosomes eventually fuse with lysosomes (pH 4.5–5.5) [18,19,20]. The resulting pH gradient in the endocytosis process has been excessively utilized to design pH-sensitive polymeric materials for NM applications [21,22]. For example, a polymeric-micelle-based carrier system was developed for selective intracellular delivery of proteasome inhibitor MG132 into cancer cells. MG132, which is a tripeptide aldehyde, was covalently bound to the block copolymer poly(ethylene glycol)(PEG)-*b*-poly(hidrazinyl-aspartamide) through an acid-labile hydrazone bond (Figure 2). This bond is stable under normal physiological conditions, but hydrolytically degradable in acidic compartments of the cell, such as late endosomes and lysosomes. Thus, it was used for the controlled release of MG132 after EPR-mediated preferential accumulation of micelles into the tumor, followed by cell internalization via endocytosis [23,24]. The same polymer platform and pH-responsive hydrazone chemistry were utilized to efficiently deliver the anthracycline drug, *viz.* epirubicin, for the treatment of glioblastoma (GBM), one of the most aggressive and difficult to treat human cancers [25,26].

Supporting the relevance of pH in physiological processes, several pathological conditions, such as cancer, inflammation, and infection, are also interrelated with pH [27]. Highly malignant solid tumors show increased interstitial tissue acidification due to enhanced glycolysis; tumor extracellular (T^ex^) pH is generally 0.3–0.7 pH units lower than the corresponding healthy tissues [28]. Additionally, tumor acidosis is heterogeneous, with the region adjacent to blood vessels having a near-neutral pH and a hypoxic region with acidic pH. Accordingly, it is quite challenging to design a bioresponsive polymer that can respond to subtle pH changes that adequately differentiates healthy tissue from tumor tissue, whilst also having an adaptive pH-responsive property. Incorporating a supramolecularly enabled dynamic covalent chemistry approach, a T^ex^ pH-triggered polymer micelle (T^ex^-micelle) type NM has been developed, loaded with desacetylvinblastine hydrazide (DAVBNH), a derivative of potent anticancer drug vinblastine. DAVBNH was conjugated to an aliphatic ketone functionalized PEG-*b*-poly(amino acid) and the hydrolytic stability of the derived hydrazone bond was efficiently tailored by exploiting the core-shell structure of polymer micelle. Effective and safe therapeutic application of T^ex^-micelle in orthotopic GBM model was achieved with significant survival benefit compared to the free DAVBNH [29]. This aliphatic ketone linker can be facilely substituted by aliphatic/aromatic aldehydes to expand the pH-responsive function of the drug conjugated polymer micelle systems. This extension of pH responsiveness can be practically utilized to achieve therapeutic outcomes depending on the biomarkers’ expression in the tumor. It was recently revealed that the therapeutic activity of pH-sensitive polymer micelles was affected by their pH-dependent intratumoral activation profile and the c-*MYC* expression of tumors, with fast-releasing polymer micelles being more effective than their slow-releasing equivalents in c-*MYC* high tumors, and vice versa [30].

In addition to small-molecule drugs, protein-carrying polymeric micelles were developed, which exploit both ion complexation between the polymer and the charged residues on the protein surface, and a pH-cleavable covalent crosslinker to construct polymer–polymer and polymer–protein crosslinks to directed the pH-triggered release of the protein payload. Towards this end, a PEG-*b*-poly(l-lysine) was reacted with carboxydimethylmaleic anhydride (CDM) to graft amine-reactive maleic anhydride units, which yields stable amide crosslinks at pH 7.4, which readily cleave at pH 6.5. The block-copolymer having 45% CDM was shown to efficiently encapsulate proteins of various sizes and isoelectric points. Additionally, a Myoglobin-loaded micelle was used to demonstrate the stability in physiological conditions, as well as the dissociation and release of functional protein at pH 6.5 [31]. Worth noting here is that by utilizing citraconic anhydride, an α-methyl derivative of maleic anhydride, the pH responsiveness can be modulated to be more in the intracellular pH range. Again, by applying *cis*-aconitic anhydride, charge density can be modulated along with intracellular pH sensitivity [32]. Similar systems have also been used for the delivery of mRNA [33].

In addition to dynamic hydrazone and pH-cleavable amide chemistry, acetals and ketals are other pH-cleavable bonds that have been broadly utilized to develop bioresponsive NMs as the pH-triggered hydrolysis of these bonds can be finely tuned by the structural variations in acetal- and ketal-based linkers [34,35,36]. For example, the hydrolysis rate of the cyclopentyl ketal at pH 5 solution was about two times slower, while that of cyclohexanone was nearly seven times slower than the acetone analog [35]. Additional to pH-responsive cleavable bonds, non-cleavable, ionizable functional groups, such as amines and carboxylic acids that can accept or release protons depending on the pH of the environment, are also frequently used in designing bioresponsive NMs, where pH-induced charge variation is the most influential factor [23,24]. Chitosan is a linear copolymer of glucosamine and *N*-acetylglucosamine, obtained by partial (>50%) *N*-deacetylation of chitin, the 2nd most prominent natural polysaccharide. Chitosan comprises ionizable amino groups with a pKa value of about 6.5, which enable its dissolution in dilute acids (pH < 6.5). Notably, a pKa of 6.5 pairs nicely with the mildly acidic environment of tumor tissues, where chitosan would be protonated, thus promoting cell adhesion via electrostatic interactions with the glycocalyx on the cell membrane. This unique character of chitosan was utilized to achieve pH-sensitive cell interactions, enabling specific drug delivery to cells in a weakly acidic microenvironment, such as a tumor [37].

As a whole, from the application point of view, pH-sensitive systems can be grossly classified as intracellular pH-sensitive, such as endo/lysosomal pH-sensitive, as well as disease tissue pH-responsive systems, such as tumor pH-responsive. However, as discussed earlier, the development of tumor microenvironment pH-sensitive NM is challenging as the pH change between tumor (pH 6.5–7) and healthy tissue or blood (pH 7.4) is narrow. On the other hand, due to the relatively large difference between physiological pH (7.4) and endo/lysosomal pH (3.5–6), designing an intracellular pH-responsive system is more viable. Nevertheless, in this case, after reaching the target site, the cell must endocytose the endo/lysosomal pH-responsive NMs and the drug should be released conditionally prior to endo/lysosomal fusion. In this case, the incorporation of targeting functions or ligands can promote selectivity towards the diseased site. In other words, the selective accumulation at the diseased site or/and selective uptake by diseased cells should be orchestrated with an intracellular release mechanism. This is especially critical for hydrolytic cargo, such as proteins and nucleic acids, where the endosomal escape mechanism is instrumental for achieving optimal benefit from intracellular-stimuli-targeted NMs.

## 3. Thermoresponsive Systems

Moderate hyperthermia up to 43 °C over an extended time is endurable by a variety of tissues with no lasting repercussions. Heat stress reduces cell survival, yet normal tissues are better equipped to handle this than malignant tissues, making hyperthermia a practical cancer treatment method that targets cancer cells and their surrounding environment, often combined with surgery and chemo-, radio-, or immunotherapy [38]. This mild hyperthermia may be induced using microwave irradiation, ultrasound, radiofrequency, infrared light, or magnetic fluid hyperthermia, making thermosensitive polymers appropriate for biological applications [39]. Poly(*N*-isopropyl acrylamide) (PNIPAAm) is a well-investigated polymer with an integral thermoresponsive character due to its temperature-dependent phase transition in aqueous solutions at 32 °C, which is the basis for subsequent phase separation and its lower critical solution temperature (LCST) behavior above 32 °C. Above the LCST, a highly hydratable PNIPAAm-coated surface rapidly becomes hydrophobic, while a crosslinked PNIPAAm hydrogel swiftly shrinks and expels its aqueous swelling solution [6,40,41]. Although 32 °C as the LCST temperature is intriguing for constructing a formulation that alters its characteristics from room temperature to physiological temperature (37 °C), attempts have been made to tailor its thermoresponsive behavior to higher temperatures (>37 °C). In this fashion, spatiotemporally controlled drug release from the polymeric drug carrier can be achieved by local heating of the pathologic site to moderate hyperthermia (43 °C) [39]. The thermoresponsive behavior of PNIPAAm can be easily modulated by copolymerization with hydrophilic or hydrophobic monomers.

Copolymerization with relevant monomers can also be utilized to introduce pH-responsive characters along with thermosensitivity, and has been utilized extensively to achieve polymer networks with desirable stimuli responsiveness. Acrylic acid (AAc) with a pKa of 4.25 is a typical pH-responsive monomer. When copolymerized with NIPAAm, it yields a pH-sensitive copolymer owing to AAc ionization as a function of pH. In crosslinked macro-, micro-, and nanogels this property can be exploited to reversibly induce pH-dependent expansion and increase osmotic pressure, due to internal charge–charge repulsion. Using a “homologous series” of acids based on the acrylic acid (AAc) backbone, and by altering the alkyl chain length of the pendant group on AAc-based comonomers (such as methacrylic acid, ethylacrylic acid, and butylacrylic acid), it was demonstrated that the pKa range can be expanded from 4.25 to 7.4, making the system responsive to more physiologically relevant pH [42]. In addition to PNIPAAm, other classes of thermoresponsive polymers are poly(2-oxazoline)s [43,44], poly(2-oxazine)s [45], poly(oligoethylene glycol acrylate)s [46], and poly(*N*-vinylcaprolactam) (PNVCL) polymers [47]. A PNVCL-based dual-stimuli-responsive copolymer system, PAA-*graft*-PNVCL, instantaneously responds to pH and temperature changes, where PAA and PNVCL perform as the pH-sensitive and thermosensitive segments, respectively [48].

Thermosensitive nanocarriers loaded with anticancer drugs are a complementary approach that promises to improve therapeutic efficacy [49]; they can be activated by local hyperthermia at the tumor site, or by hyperthermia to stimulate drug release, or a synergistic combination to kill cancer cells [50]. Though it should be considered that hyperthermia, which can be local or whole-body hyperthermia, is associated with adverse effects such as pain at the site, infection, bleeding, blood clots, swelling, burns, and damage to the skin, muscles, and nerves near the treated area; furthermore, nausea, vomiting, and diarrhea can occur from whole-body hyperthermia. To reiterate, hyperthermia is still an experimental procedure requiring specialized equipment and a qualified doctor and treatment team. As a result, hyperthermia is not available at all cancer treatment clinics. Accordingly, the inherent limitations or challenges associated with hyperthermia must be considered when designing thermoresponsive materials for NM purposes. Although challenging to achieve, thermoresponsive polymers, where the phase separation/transition can be triggered with very mild hyperthermia (perhaps merely a few degrees higher than the physiological temperature), and minimal exposure time would be advantageous. In this case, an additional stimulus, such as pH, can be coordinated to achieve that delicate balance.

## 4. Enzyme-Responsive Systems

Enzymes are essential in all biological and metabolic processes. Most enzymes catalyze chemical reactions under mild temperature, pressure, or pH, compatible with the biological milieu, where many conventional chemical reactions fail. Dysregulation of enzyme expression and activity is characteristic of many diseases and a potential biological trigger in therapeutics. Enzymes can also possess high substrate selectivity, enabling precise, complex, biologically-inspired chemical reactions that can have added benefits for designing modern therapeutics.

Many enzyme classes are overexpressed in tumor microenvironments, such as proteases, lipases, and phosphatases, which serve as potential targets to develop bioresponsive polymeric materials for cancer NM; among proteases, cathepsins [51] and matrix metalloproteinases [52] are the most frequently used stimuli [53]. Cathepsin B (CTSB) is a lysosomal protease, and several human cancers are reported to have elevated expression of CTSB, which has also been suggested as a potential cancer biomarker. CTSB hydrolyzes peptides comprise Leu-Leu, Arg-Arg, Ala-Leu, Phe-Arg, Phe-Lys, Gly-Phe-Leu-Gly, and Ala-Leu-Ala-Leu, exclusively; of these peptides, Gly-Phe-Leu-Gly (GFLG) is the most commonly used substrate. Several effective CTSB-responsive antitumor drug delivery systems have been developed by capitalizing on differences in CTSB concentrations and activities between tumor and healthy tissues. To improve the therapeutic efficacy of a widely used anticancer drug gemcitabine (GEM), a prodrug was designed where GEM is conjugated with a dendritic poly(*N*-(2-hydroxypropyl) methacrylamide) (polyHPMA) copolymer via the CTSB-cleavable tetrapeptide linker GFLG. The dendritic architecture of the prodrug (polyHPMA-GEM) aggregates and forms stable nanoscale systems of high molecular weight (HMW, 168 kDa) that biodegrade into kidney-excretable segments of low molecular weight (LMW, 29 kDa), and demonstrates enzyme-responsive drug release features in the presence of CTSB [54].

The accessibility of the substrate to the target enzyme is a crucial consideration when designing enzyme-responsive biomaterials; especially for systems with supramolecular nanoassemblies, accessibility can be limited or even restricted depending on the positioning of the enzyme-sensitive bonds or linkers within the nanostructure. Accordingly, it is important to gain critical insight into the disassembly or degradation process of enzyme-sensitive supramolecular structures. This critical understanding of enzyme responsiveness on disassembly or degradation process was elaborately studied using CTSB-sensitive supramolecular peptide amphiphile (PA) nanofibers, where the exopeptidase and endopeptidase features of cathepsin B were investigated to gain a better perception of its enzymatic degradation process. Experimental results revealed that assembled PA nanofibers degraded by CTSB through a surface erosion mechanism as only the surface amino acid residues were cleaved due to the easy accessibility. Understandably, the erosion of the PA nanofibers was caused by CTSB degradation on the assembled nanofibers rather than on monomers dissociated from the assemblies. Additionally, the number of cleaved residues and degradation efficiency were found to be inversely related to the internal viscosity of the PA nanofibers; particularly, high internal viscosity from well-packed β-sheet conformation of PA molecules significantly reduced the percentage of degradation and the number of cleaved C-terminal amino acid residues [55].

## 5. ROS-Sensitive Systems

Reactive oxygen species (ROS) refer to a series of chemical molecular oxygen species, including singlet oxygen (^1^O_2_), hydrogen peroxide (H_2_O_2_), superoxide anion (O_2_^•−^), and hydroxyl radical (^•^OH)**.** These species serve as cell signaling molecules for normal biologic processes [56,57]. However, the production of ROS can cause damage to a variety of cellular organelles and functions, eventually disrupting normal physiology. When ROS production outweighs antioxidant defenses, *viz.* the ROS scavengers, cells experience oxidative stress, which is linked to many pathophysiologic conditions, including cancer. High levels of ROS, generated from continuous aerobic glycolysis followed by pyruvate oxidation in mitochondria (the Warburg effect), enhance receptor and oncogene activity, and activate growth-factor-dependent pathways, or oxidizing enzymes induce genetic instability; which are all features of cancer [58]. Other diseases, such as rheumatoid arthritis, diabetes mellitus, inflammation, cardiovascular diseases, atherosclerosis, obesity, obesity-related comorbidities, and neurological disorders, are also associated with oxidative stress. Correspondingly, the abnormal ROS levels in many pathologic conditions, particularly cancer, are an appealing feature to design and develop responsive polymeric NMs for selective intervention in diseased tissues [59,60]. However, it is challenging to develop polymers with enough responsiveness to distinguish pathological ROS levels from healthy ROS levels. This challenge was delicately addressed by incorporating a chemical amplification strategy in designing a ROS-responsive polymer (ROS-ARP) [59]. The designed polymer is composed of three monomers, each possessing different functionalities, where ketal moieties trigger the degradation of the polymer upon hydrolysis of the ketal groups on the polymer backbone. Utilizing a logic gate system, the poor hydrolytic stability issue of ketal functions was managed. The logic gate system is based on the different ketal hydrolysis rates, and hinges on the hydrophilicity of the polymers where the ketal groups are incorporated in. Two ROS-sensitive functional groups were introduced into the polymer to control the hydrolysis rate of the ketals. To begin, a thioether group is introduced via Michael addition of a bisthiol to the bisacrylamide ketal, thus inducing polymerization whilst incorporating a ROS-sensitive hydrophilicity switch (Figure 3). By interacting with ROS such as H_2_O_2_ and HOCl, the thioethers are first oxidized to sulfoxides and subsequently to sulfones. The oxidation of the thioethers in the polymer backbone affects the polarity and hydrophilicity of the polymer, and consequently the hydrolysis rate of the ketals. Another ROS-sensitive functional group is hydroxymethyl-phenylboronic acid pinacol ester (BE), which is well documented to be quantitatively cleaved with excellent selectivity and sensitivity by H_2_O_2_. This functional group was coupled to the carboxylic acid of an acryloyl modified lysine to produce an acid-masked H_2_O_2_-responsive group. The generation of COOH groups triggered by H_2_O_2_ amplifies the ROS signal and catalyzes the hydrolysis of the ketals in the polymer backbone. It was confirmed that the ROS-ARP degraded over 17 times faster than its control polymer without a chemical amplification strategy. While the design of polymers to realize ROS-triggered rapid degradation using chemical amplification is intriguing, the therapeutic application of this system is yet to be confirmed.

In another example, a theranostic nanoplatform with serial ROS responsiveness and two-photon aggregation-induced emission (AIE) bioimaging has been constructed for dimensional diagnosis and accurate inflammation therapy [61]. A commonly used glucocorticoid drug known as Prednisolone (Pred) was bridged to a two-photon fluorophore (TP) via a ROS-sensitive bond to form a diagnosis–therapy compound TPP, which was then encapsulated by the amphipathic poly(2-methacryloyloxyethyl phosphorylcholine) (PMPC)-poly(2-methylthio ethanol methacrylate) (PMEMA) polymer, PMM, that self-assembles into the core−shell-structured micelles (TPP@PMM). The specific pathophysiological condition of the inflammation site, where edematous tissue and leaky vasculature allow the TPP@PMM micelle to extravasate and accumulate at the inflammatory tissue, facilitates the selective release of Pred through the serial response to the local overexpressed ROS.

## 6. Glutathione/Reductive Environment-Responsive Systems

Following our discussion on ROS and the associated delicate balance between ROS production and ROS scavenging, it is worth highlighting the significance of reduced glutathione (GSH) in maintaining this critical balance [62]. GSH is a linear tripeptide of l-glutamine, l-cysteine, and glycine, and is one of the most abundant and significant scavengers of ROS in eukaryotic cells. During the ROS scavenging process, GSH is converted into oxidized glutathione (GSSG), the corresponding disulfide of GSH. Nevertheless, glutathione predominantly exists in its reduced form as the glutathione reductase enzyme continually reduces the disulfide bond of GSSG. Accordingly, the GSH:GSSG ratio is considered a critical cellular oxidative stress biomarker; in a resting cell the molar GSH:GSSG ratio exceeds 100:1. While in the event of oxidative stress, this ratio has been revealed to decrease to values of 10:1, and even 1:1 [63]. Under normal physiological conditions, GSH plays a crucial role in maintaining the intracellular reductive environment, with concentrations usually ranging from 1 to 10 mM, whereas extracellular (in plasma) values are one to three orders of magnitude lower [62,64]. This extreme intracellular and extracellular concentration gradient of GSH is a particularly attractive phenomenon for engineering materials responsive to this difference in reductive environment. For example, a polymeric delivery system that can withstand the mild extracellular reductive environment disintegrates intracellularly upon exposure to the harsh reductive environment, releasing the cargo inside the cell. The disulfide bridge, in which the intracellular GSH content is exploited to stimulate thiol–disulfide exchange, is one of the most frequently applied dynamic covalent bonding for accomplishing this goal. In these cases, the disulfide linkage is mainly used as a crosslinking tool. Disulfide conjugation, also known as thiol–thiol coupling, has been a tremendously popular conjugation strategy among researchers due to its orthogonality and ease of installation onto polymers. Kataoka et al., utilized disulfide crosslinking chemistry comprehensively in polyplex micelle platforms to deliver a wide range of cargo molecules, such as plasmid DNA [65], siRNA [66], mRNA [67], and macrocycle phthalocyanine [68]. It is noteworthy that while disulfide crosslinking improves the stability of the polyplex micelle, and provides higher cargo protection during circulation, it is essential to precisely tune the thiol–thiol crosslinking process to achieve optimal intracellular delivery [67]. In addition to polymer–polymer crosslinking to stabilize polyplex micelle core, disulfide linkage has also been adopted to detach PEG chains from a polyplex micelle for nonviral gene delivery. P(Asp(DET)) homopolymer-derived polyplexes demonstrate a higher transfection efficiency compared to PEG-P(Asp(DET)) micelles, especially at low cation/anion ratios [69], indicating that the PEG shell of the polyplex is somehow interfering with the transfection, a phenomenon often termed the “PEG dilemma”. To address this issue a catiomer, PEG-SS-P(Asp(DET)), was designed, where an insertion of a disulfide linkage between PEG and the polycation segment was introduced to trigger PEG detachment under the intracellular reductive environment. Furthermore, a cationic segment based on P(Asp(DET)) was used as a buffering moiety, thus inducing endosomal escape with minimal cytotoxicity [70]. In a recent study, this disulfide chemistry for PEG-detachment was employed to deliver a programmed death-ligand 1 antibody (aPD-L1) into a GBM site. For this purpose, a aPD-L1-PEG-glucose conjugate was synthesized, whereby the glucosylated PEG enabled active transportation of the aPD-L1 across the vasculature of GBM through recognition by the glucose transporter 1 (GLUT-1). In this manner, an active immune response against GBM could be facilitated after PEG detachment, thus recovering the native aPD-L1 at the target side [71]. 

Although disulfide linkage has been applied in the design of many drug and gene delivery systems [72,73], it is well-recognized that the reaction kinetics of disulfides is not readily tunable [74]. Hence, a thioester-based strategy was adopted, where the reactivity can be conveniently modulated by choosing the appropriate steric environment around the thioester. Utilizing this strategy, the activity of a tumor-necrosis-factor-related apoptosis-inducing ligand (TRAIL), was reversibly PEGylated to improve TRAIL’s in vivo therapeutic efficacy (Figure 4). In this study, the rate of traceless dePEGylation of proteins was tuned by varying the steric hindrance around the thioester moiety [74].

## 7. ATP-Responsive Systems

Similar to the marked difference in intracellular and extracellular GSH concentration, adenosine triphosphate (ATP), another biologically significant molecule, also represents a striking intra/extracellular concentration gradient. ATP, a nucleotide consisting of an adenine base attached to a ribose sugar, which is attached to three phosphate groups, is the molecular unit of intracellular energy currency. ATP is produced by actively growing cells for short-term energy storage and transfer, where the inherent energy of ATP comes from high-energy phosphate bonds. Adenosine diphosphate (ADP) and adenosine monophosphate (AMP) are precursors of ATP, and together, the three maintain the cellular adenylates. ADP and AMP rotate between the sites where high-energy phosphate bonds are added to form ATP and areas where the phosphate bonds are broken to transfer energy to a metabolic process. ATP also operates as a neurotransmitter in the peripheral and central nervous systems, acting as a cofactor for signal transduction processes involving several kinases and adenyl cyclase. ATP is stored in very high amounts within the cells, in the range of 3–10 mM, with a typical ratio of ATP/ADP of approximately 1000, whereas the extracellular concentration is considerably lower, the estimated concentration of 10–100 nM.

Utilizing this remarkable intra/extracellular ATP concentration gradient, a polyplex micelle type nanocarrier with ATP-responsive inner core crosslinking strategy was designed to efficiently deliver messenger RNA (mRNA) with a specific focus on shielding the mRNA from enzymatic degradation and delivering mRNA specifically inside the cytosol for coherent mRNA translation [75]. In ATP-responsive crosslinking chemistry, the well-studied phenyl boronate ester bond generated from the spontaneous coupling of phenylboronic acid (PBA) and polyols installed onto PEG-*b*-polycation was utilized (Figure 5). It is well-known that PBAs readily form boronic esters with 1,2- and 1,3-*cis*-diol compounds, including those found in carbohydrates, such as glucose or ribose in a reversible manner. This phenyl-boronate-ester-based dynamic covalent chemistry has been previously utilized for the intracellular delivery of siRNA using polyion complex micelles [76,77,78]. Critically, in the case of polyplex-micelle-derived mRNA carrier design, the ratios of phenyl boronate ester crosslinkers, and the structure and the protonation degree of amino groups in the polycation segment of block copolymers, were found to have a significant impact towards maximizing protein expression in cultured cells. This observation was attributed to the careful balance between the robustness in the biological milieu, and the ATP-responsive mRNA release in the cytosol. Furthermore, to achieve robust blood circulation following intravenous administration of mRNA-loaded polyplex micelles, cholesterol moieties were installed onto both the mRNA and ω-end of the block copolymer, which improves the stability of the polyplex micelle through the stacking of cholesterol components.

In another example, the same polymer platform was used for intracellular delivery of enzymes [79]. In this work, mPEG-*b*-poly(2-((2-aminoethyl)amino)ethylaspatamide) polymer was modified with PBA to assemble enzymes, such as glucose oxidase, by electrostatic interactions into enzyme nanoclusters. The assembled polymer–enzyme nanoclusters have a relatively low enzyme activity, thus avoiding non-specific catalysis during blood circulation. When the nanoclusters reach the tumor site and undergo cellular internalization, boronate ester on the nanoclusters exchange with the diols on the ribose moieties of intracellular ATP, thus accelerating the reversal of charge and hydrophobic nature of the polymer, ultimately causing the disassembly of the nanoclusters and the release of the enzymes in the cytosol.

## 8. Combination of Stimuli

Accumulating data suggesting that NM responsive to a combination of stimuli can provide superior spatiotemporal control of drug action at the diseased site, has triggered extensive research into multiresponsive NM for the targeted therapy of a variety of disorders, including cancer, diabetes, and inflammatory and CNS diseases. An active targeting nanoparticle with pH- and ROS-responsive features was engineered, aimed at precision delivery of rapamycin, which is a drug candidate for vascular inflammation, relying on the intrinsic acidosis and oxidative stress associated with inflammation [80]. In another instance, a dual pH–redox-responsive polymeric nanomicelle capable of delivering therapeutically doses of 3D6 antibody fragments (3D6-Fab) into the brain parenchyma was designed, in order to prevent Alzheimer’s disease (AD)-associated amyloid-beta aggregation (Figure 6). A charge-converting 3D6-Fab was used to facilitate complexation with reductive–sensitive cationic polymer PEG-poly(l-lysine) via pH-sensitive citraconylation. A glucose-decorated nanomicelle surface was incorporated to enable effective interaction with GLUT-1 transmembrane transporters, thus achieving brain targeting. The 3D6-Fab-loaded micelles remained stable during systemic circulation, but disassembled in a stepwise manner in the acidic endosomal environment of brain cells and the reductive brain parenchyma [81]. AD is a progressive and irreversible brain disorder that slowly degrades memory and cognitive function, accounting for 60–70% of dementia cases. Given the seriousness of the disease and the continual increase in patient numbers, developing effective therapies to treat AD has become urgent. Current antibody (Ab)-based therapeutics are promising for the treatment of AD. Acuranumab (brand name Aduhelm^TM^) is a monoclonal antibody that targets amyloid-beta and prevents amyloid buildup. Its potential was recognized by the Food and Drug Administration (FDA) in June 2021, and gained accelerated approval for the treatment of AD. However, brain delivery of Abs in therapeutic relevant concentrations remains a challenge due to the blood–brain barrier (BBB). On this occasion, technologies that improve the bioavailability of Ab-based treatment platforms within the central nervous system parenchyma, such as the dual pH–redox-responsive glucose decorated polymeric nanomicelle loaded with 3D6-Fab, are expected to have tremendous clinical benefits. This example also signifies how strategic biomaterial engineering can address a serious medical issue in the context of bioresponsiveness.

While nanoformulations sensitive to a combination of stimuli have the potential to achieve superior spatiotemporal activation profiles over responsiveness to a single stimulus, arduous synthesis or complex engineering processes can become a challenge for clinical translation. Additionally, the lack of optimization for each stimulus and poor overall characterization of the nanocarrier also undermines the potential of multifunctional platforms. Importantly, multifunctionality does not always offer significant therapeutic benefits. Accordingly, it is essential to consider the balance between added complexity in material design, i.e., technology to achieve multifunctionality, and the benefits from each added component of NM.

## 9. Summary and Future Perspectives

Bioresponsive materials are an integral tool for realizing NM-based targeted therapy, an advanced version of Paul Ehrlich’s magic bullet concept that precisely targets diseased tissue or cells, sparing the healthy counterparts. We discussed this perception with a few specific examples and literature reviews that highlight the enormous research effort towards realizing targeted NM utilizing bioresponsive systems with diverse biological perspectives. Although these colossal research efforts produced some success with few stimuli-responsive systems even reaching to clinical trials (Table 1), Paul Ehrlich’s magic bullet concept concerning NM is yet to be realized with full potential, including the integral bioresponsive feature. One of the critical reasons for this apparent deficit is probably the complex nature of biological systems in general that becomes more convoluted in the event of disease. Another critical contemplation is that NMs designed to administer systemically must maintain their stability/integrity during circulation. In other words, the amount of payload associated with NM over time following administrations is a critical consideration in designing nanoformulations. Again, NM must release its therapeutic payload after reaching the target site. The rate at which the cargo is released is critical, as it determines the amount of active drug available over time to maintain optimal “drug–target kinetics”, a term that defines drug–target complex formation and breakdown [82,83,84,85]. This drug–target kinetics is an especially critical factor for molecularly targeted therapeutics due to the reversible binding nature of this drug class. “Drug–target kinetics” and associated “drug–target residence time” [86,87] are also decisive factors for drug or drug-loaded nanocarrier development against brain diseases, as in most cases, only a limited amount of drugs can reach the brain parenchyma due to BBB related hurdle [85]. Drug release kinetics may also change the toxicity profile of NM. For example, a cytotoxic-drug-loaded NM with fast release or activation kinetics might go through burst release during the elimination process, exposing the excretory organs to a significant amount of toxic drug and thus limiting the dose threshold [29]. Regarding the bioresponsiveness of NM, specifically, the subtle differentiation between healthy and diseased tissue poses an additional challenge to maintain a delicate balance between two opposite dynamics, drug encapsulation vs. drug release, assembly vs. disassembly, or association vs. dissociation. Accordingly, some selectivity could be achieved, but not the expected specificity. However, this balanced situation is still acceptable if selectivity is retained to a level of maximal therapeutic benefit without substantial adverse effects. In this context, defining the exact selectivity for diseased tissue over healthy ones is critical, in addition to precise evaluation techniques to quantify this selectivity. For a stimuli-responsive system, the responsivity to certain stimuli is often evaluated in test-tube conditions that mimic the stimuli environment in question. For example, buffered aqueous solutions with different pH levels are used for pH-sensitive systems, and for glutathione-responsive systems, high and low glutathione concentrations are used. These systems are too simple and do not mimic the complex biological situation. Accordingly, overappraisal or underestimation of responsiveness is probable. Additionally, these release conditions are generally static and rarely reflects the gradual shift, overlapping or heterogeneity of biological-stimuli conditions. Recently, the reflection of spatial heterogeneity in terms of tumor acidosis was demonstrated in pH-dependent drug release kinetics that gradually accelerates as the pH increases. In this example, the release media was changed in a pH-descending manner, which confirmed the presence of a progressive release profile of the tumor extracellular pH-triggered polymer micelle (T^ex^-micelle) [29]. Another dilemma that is often encountered in designing release experiments from bioresponsive NM is whether to adopt sink conditions or non-sink conditions. Although a sink condition is more relevant to the biological situation, it is often challenging to achieve due to the cumbersome experimental settings and limitations in analytical sensitivity; from these considerations the non-sink condition is still justifiable under appropriate experimental conditions. However, in this case, the limitation of non-sink conditions in terms of bioequivalence should be reflected during the interpretation of result.

Evaluating stimuli sensitivity in the test tube under appropriately selected conditions [84] is valuable during the initial development stage of bioresponsive systems; however, it is also critical to confirm the true nature of responsivity in the biological setting. Accordingly, it is vital to develop effective methods to evaluate the spatiotemporal activation profile of NMs and to measure the concentration and distribution of the active payload, both on and off target. A recently reported in vivo release experiment investigated the spatiotemporal drug release from pH-sensitive and drug-loaded polymer micelles that were intratumorally injected into subcutaneously implanted tumors [29]. The tumors were subsequently collected from where the conjugated drug (quantified after acid hydrolysis of the sample), released drug (quantified in the sample without any acid treatment), and total drug (released + conjugated) within the tumor mass were quantified using LC/MS. Acid treatment was performed in order to cleave the drug from the drug–polymer conjugate prior to quantification. While in this case, only the total amount of drug (released or conjugated) was quantified, in another example, the powerful matrix-assisted laser desorption/ionization mass spectrometry imaging (MALDI-MSI) technique was effectively used to analyze in vivo spatiotemporal distribution of released drugs from pH-sensitive NMs in the tumor sections [30]. In addition to precise quantification of NM-mediated delivery efficacy in diseased tissue, the intracellular fate of payloads also needs to be determined with accuracy, especially where the target is a specific organelle of the cell. Detection of cytosolic delivery of fluorescently labeled siRNA in lipid nanoparticles (LNP) was accomplished by a confocal microscopy-based method that enabled detection and quantification of subnanomolar levels of cytosolic siRNA from individual release events with measures of quantitation confidence for each event. Additionally, single-cell kinetics of siRNA-mediated knockdown in cells expressing destabilized eGFP showed a dose–response connection for knockdown induction, depth, and duration in the range from several hundred to thousands of cytosolic siRNA molecules [88]. These examples signify the enormous possibilities of the already available techniques to perform in vivo release/activation experiments on bioresponsive NMs, and reinforce further development of experimental protocols that is expected to assist smooth translation.

Thus far, we have explored and discussed the critical challenges of the current research trend to realize the full potential of bioresponsive polymer for NM application. To progress further, we must consider the future direction of the field. In this era of precision medicine that signifies a patient-centric approach, where identifying specific predictive biomarkers drives therapeutic choices for individuals, NM research must take a similar trajectory to keep up with the transformative momentum of translational research. In this direction, bioresponsiveness must also be realized from the perspective of interpatient variability; meaning further fine-tuning or adaptability of the chemistry and material design that dictates the sensory function of NM.

## Figures and Tables

**Figure 1 polymers-14-03659-f001:**
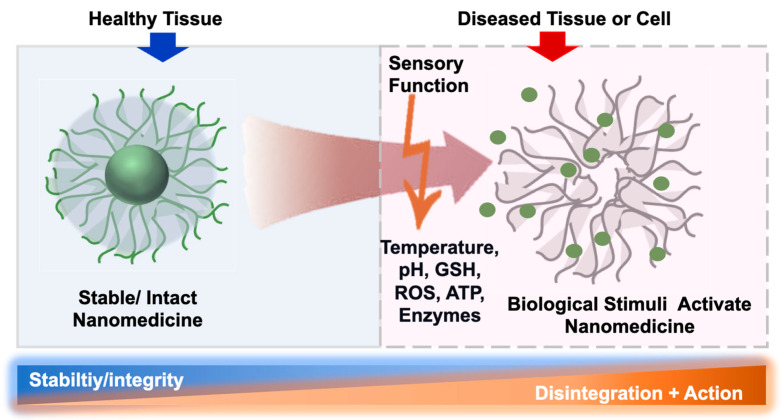
Schematic representation of nanomedicine’s bioresponsive and sensory functions.

**Figure 2 polymers-14-03659-f002:**
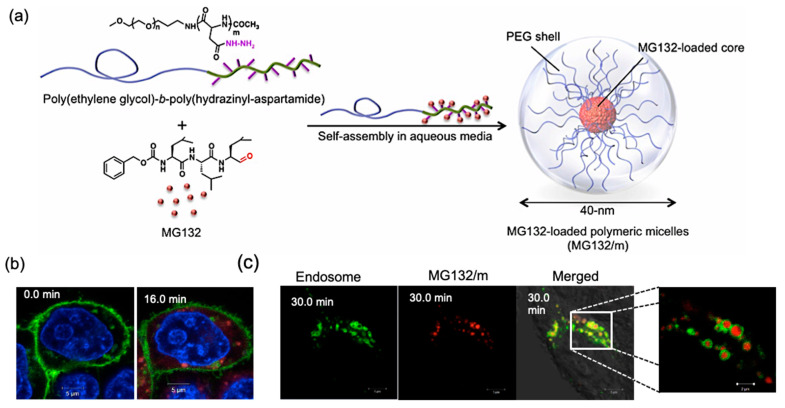
Proteasome inhibitor with MG132-loaded pH-sensitive polymeric micelles. (**a**) Proteasome inhibitor MG132 was conjugated to the hydrazide moieties of PEG-*b*-poly(hidrazinyl-aspartamide) copolymer, and this polymer subsequently self-assembled in aqueous media to form polymer micelles with size of around 40 nm. (**b**) In vitro CLSM images of dissociation of BODIPY TR-labeled MG132/m (red) in HeLa-luc cells. Cell membrane was tagged with GFP (green), and the nucleus with Hoechst (blue). Absence of colocalization of micelles and cell membrane confirms the disintegration of the micelles only in the intracellular space. (**c**) In vitro CLSM images of the dissociation of BODIPY TR-labeled MG132/m (red) in late-endosomal compartments, which were marked with GFP (green); enlarged section is shown in the flap. Reprinted with permission from [23]. Copyright Elsevier 2014.

**Figure 3 polymers-14-03659-f003:**
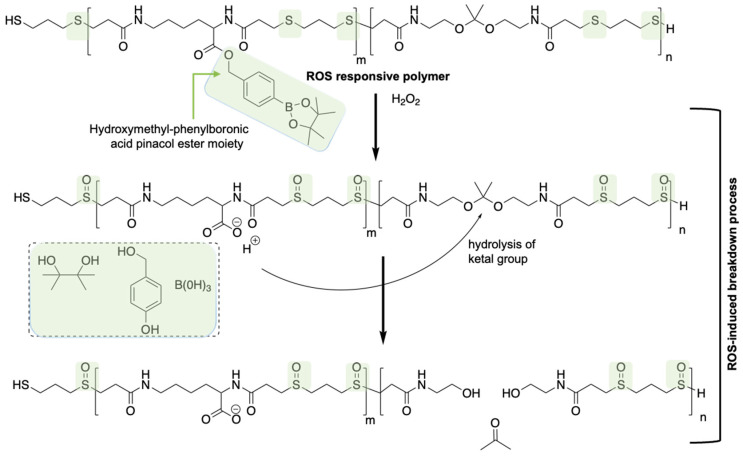
Schematic representation of H_2_O_2_-responsive ROS-ARP degradation mechanism. ROS-responsive moieties are highlighted in green. Recreated from [59].

**Figure 4 polymers-14-03659-f004:**
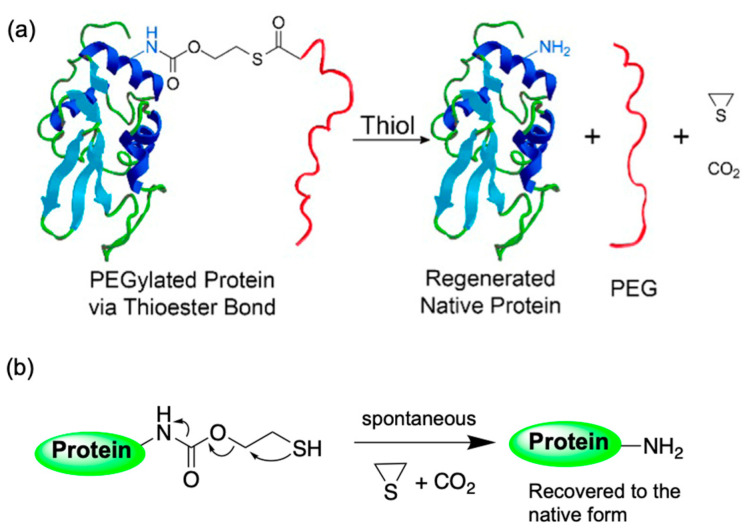
Example of glutathione/reductive environment-sensitive traceless PEG detachment. (**a**) Schematic representation of thioester-mediated, traceless, reversible PEGylation of protein. (**b**) Chemistry involved in protein regeneration after PEG detachment via thioester-cleavage-generated thiol group triggers an intramolecular reaction to generate a three-membered thiirane and CO_2_, rescuing the protein in its native form. Reprinted with permission from [74]. Copyright ACS 2013.

**Figure 5 polymers-14-03659-f005:**
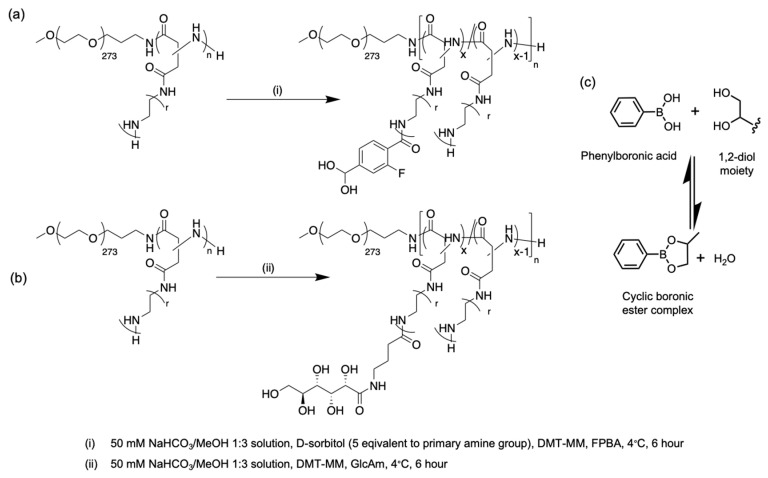
Scheme for installing phenylboronic acid and polyols on to PEG-*b*-polycations and representation of subsequent boronic ester formation/breakdown. Reaction conditions for introducing (**a**) 4-carboxy-3-fluorophenylboronic acid (FPBA) and (**b**) *N*-gluconamide (GlcAm) as polyol segments on polymers. (**c**) Representation of equilibrium between, formation, and breakdown of boronic ester. Recreated from [75].

**Figure 6 polymers-14-03659-f006:**
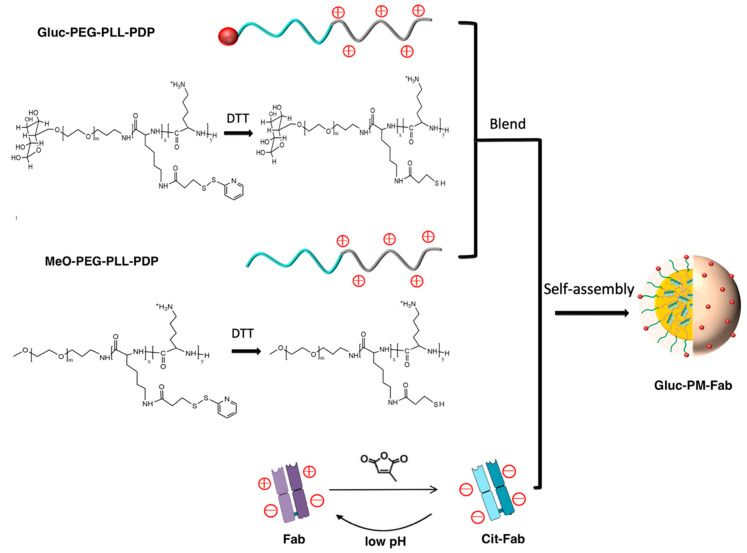
Graphical representation of the design of a dual pH–redox-sensitive, Fab-encapsulated, glucosylated polymeric nanomicelle (Gluc-PM-Fab) development *via* the assembly of negatively charged Cit-Fab and the positively charged block copolymers. Reprinted with permission from [81]. Copyright ACS 2020.

**Table 1 polymers-14-03659-t001:** Bio-Responsive Polymer-based Nanomedicines that are in the clinical trial.

Trade Name	Stimuli	Nature of the Responsiveness	Nanomedicine Platform	Active Agent	Therapeutic Application	Phase	Clinical Trail Number
NC6300	pH	pH-sensitive drug release	Polymeric micelle	Epirubicin	Soft Tissue Sarcoma Metastatic Sarcoma	1 and II	NCT03168061
ThermoDox® (Celsion)	Temperature	thermosensitive -mild hypothermia initiates drug release	Liposome	Doxorubicin	For a variety of cancers, such as pancreatic, breast, colon, liver, sarcoma and so on.	I to III	NCT04852367, NCT04791228, NCT02536183 and others
LiPlaCis	Enzyme	secretory phospholipase A_2_ (sPLA_2_) sensitive drug release	Liposome	Cisplatin	Advanced or Refractory Solid Tumors, Metastatic Breast Cancer, Prostate Cancer and Skin Cancer	I and II	NCT01861496

## Data Availability

Not applicable.

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
