# Peer review of "Bioresponsive Polymers for Nanomedicine—Expectations and Reality!"

_polymers, 2022, doi:10.3390/polym14173659_

Round 1

Reviewer 1 Report

The figures require improvement, since the ones provided look like copy and paste from other place, thus please include the original ones with improved quality.

If you will not include any acknowledgments, rows 592-594, please do not include this in the manuscript.

Author Response

The figures require improvement, since the ones provided look like copy and paste from other place, thus please include the original ones with improved quality.
Thank you very much for the helpful comment. The figures have been recreated.
If you will not include any acknowledgments, rows 592-594, please do not include this in the manuscript.
I understand this section will ultimately be removed.

Reviewer 2 Report

This is a very good review article dealing with the use of bio-responsive polymers in nanomedicine. The description of this approach as an advanced version of the Paul Ehlrich's magical bullet was very impressive. I recommend the acceptance of this review after performing the following:

1- Addition of a table that compares the advantages and limitations of all the methods and listing the performed studies and citing the releavant references would be very helpful for the researchers to proceed in this field.

2- The authors should add 2 more important sections for the electrical-responsive (conductive) polymers and also the light sensitive polymers (For example sensitive to the near-infrared (NIR)/ The authors can use the following articles:

https://doi.org/10.1016/j.jddst.2018.07.002 And International Journal of Nanomedicine 2020, 15:2605-2615

Author Response

This is a very good review article dealing with the use of bio-responsive polymers in nanomedicine. The description of this approach as an advanced version of the Paul Ehlrich's magical bullet was very impressive. I recommend the acceptance of this review after performing the following:

Thank you very much for reviewing our manuscript and also for your thoughtful appreciation of our work.

  • Addition of a table that compares the advantages and limitations of all the methods and listing the performed studies and citing the releavant references would be very helpful for the researchers to proceed in this field.

This is a very good suggestion. Considering the design of the article, we decided to add a table that highlights the clinical application potential of bio-responsive nanomedicines.    

  • authors should add 2 more important sections for the electrical-responsive (conductive) polymers and also the light sensitive polymers (For example sensitive to the near-infrared (NIR)/ The authors can use the following articles:

https://doi.org/10.1016/j.jddst.2018.07.002 And International Journal of Nanomedicine 2020, 15:2605-2615

Thank you very much for your comment. In this review, we aimed to highlight mainly endogenous or biological stimuli. Accordingly, we think a description of electrical-responsive and light-sensitive polymers would not fit with the overall tone of the review.